# Repurposing Atovaquone as a Therapeutic against Acute Myeloid Leukemia (AML): Combination with Conventional Chemotherapy Is Feasible and Well Tolerated

**DOI:** 10.3390/cancers15041344

**Published:** 2023-02-20

**Authors:** Alexandra McLean Stevens, Eric S. Schafer, Minhua Li, Maci Terrell, Raushan Rashid, Hana Paek, Melanie B. Bernhardt, Allison Weisnicht, Wesley T. Smith, Noah J. Keogh, Michelle C. Alozie, Hailey H. Oviedo, Alan K. Gonzalez, Tamilini Ilangovan, Alicia Mangubat-Medina, Haopei Wang, Eunji Jo, Cara A. Rabik, Claire Bocchini, Susan Hilsenbeck, Zachary T. Ball, Todd M. Cooper, Michele S. Redell

**Affiliations:** 1Department of Pediatric Hematology/Oncology, Texas Children’s Cancer Center, Baylor College of Medicine, Houston, TX 77030, USA; 2Development, Disease Models & Therapeutics Graduate Program, Baylor College of Medicine, Houston, TX 77030, USA; 3Department of Pharmacy, Texas Children’s Hospital, Baylor College of Medicine, Houston, TX 77030, USA; 4Department of Pediatrics, Baylor College of Medicine, Houston, TX 77030, USA; 5Department of Chemistry, Rice University, Houston, TX 77005, USA; 6Duncan Comprehensive Cancer Center, Baylor College of Medicine, Houston, TX 77030, USA; 7The Sidney Kimmel Comprehensive Cancer Center, Johns Hopkins University School of Medicine, Baltimore, MD 21231, USA; 8Department of Pediatric Infectious Diseases, Baylor College of Medicine, Houston, TX 77030, USA; 9Cancer and Blood Disorders Center, Seattle Children’s Hospital, Seattle, WA 98105, USA

**Keywords:** pediatric, oxidative phosphorylation, metabolism, xenograft, patient-derived, oxygen consumption rate, pneumocystis jiroveci pneumonia

## Abstract

**Simple Summary:**

Novel well-tolerated agents are urgently needed to improve outcomes for children with acute myeloid leukemia. Atovaquone is an anti-infective agent which can be used to prevent and treat a type of pneumonia that all children with acute myeloid leukemia require prophylaxis against. In addition, atovaquone has been shown to have anti-leukemic effects. In the clinical trial described here, atovaquone is tolerated well during intensive chemotherapy with no attributable adverse events. However, perhaps due to side effects from intensive chemotherapy, plasma concentrations of atovaquone were lower than expected. Thus, for patients getting intensive leukemia-directed therapy, atovaquone plasma concentrations should be followed. Additionally, embedded correlative biology studies demonstrated that atovaquone produced anti-leukemia effects in patient samples in vitro and in patient-derived xenograft models. Our results support further study of atovaquone and other agents that target the dysregulated metabolism of acute myeloid leukemia cells.

**Abstract:**

Survival of pediatric AML remains poor despite maximized myelosuppressive therapy. The *pneumocystis jiroveci pneumonia* (PJP)-treating medication atovaquone (AQ) suppresses oxidative phosphorylation (OXPHOS) and reduces AML burden in patient-derived xenograft (PDX) mouse models, making it an ideal concomitant AML therapy. Poor palatability and limited product formulations have historically limited routine use of AQ in pediatric AML patients. Patients with de novo AML were enrolled at two hospitals. Daily AQ at established PJP dosing was combined with standard AML therapy, based on the Medical Research Council backbone. AQ compliance, adverse events (AEs), ease of administration score (scale: 1 (very difficult)-5 (very easy)) and blood/marrow pharmacokinetics (PK) were collected during Induction 1. Correlative studies assessed AQ-induced apoptosis and effects on OXPHOS. PDX models were treated with AQ. A total of 26 patients enrolled (ages 7.2 months–19.7 years, median 12 years); 24 were evaluable. A total of 14 (58%) and 19 (79%) evaluable patients achieved plasma concentrations above the known anti-leukemia concentration (>10 µM) by day 11 and at the end of Induction, respectively. Seven (29%) patients achieved adequate concentrations for PJP prophylaxis (>40 µM). Mean ease of administration score was 3.8. Correlative studies with AQ in patient samples demonstrated robust apoptosis, OXPHOS suppression, and prolonged survival in PDX models. Combining AQ with chemotherapy for AML appears feasible and safe in pediatric patients during Induction 1 and shows single-agent anti-leukemic effects in PDX models. AQ appears to be an ideal concomitant AML therapeutic but may require intra-patient dose adjustment to achieve concentrations sufficient for PJP prophylaxis.

## 1. Introduction

Acute myeloid leukemia (AML) is a hematologic malignancy for which steady improvement in outcomes from further intensification of standard cytotoxic chemotherapy appears to have plateaued [1]. Despite advances in molecular characterization and risk stratification, AML is associated with high rates of chemotherapy resistance, relapse, and therapy-associated death. Thus, novel additive therapies with minimal innate and overlapping toxicity profiles are urgently needed [2,3,4]. Chemoresistance may be mediated by protective influences of the bone marrow microenvironment which ultimately allow for relapse of disease [5].

One resistance mechanism for adult AML stem cell survival is a dependence on alternative metabolic pathways such as oxidative phosphorylation (OXPHOS) [6,7,8]. Chemoresistant adult AML cells are OXPHOS dependent, and the BCL inhibitor, venetoclax, which also suppresses OXPHOS, is highly effective in both adults and children with AML [9,10,11,12,13,14,15]. Our data suggest a similar dependency on OXPHOS in pediatric AML, highlighting a compelling rationale to study medications that suppress OXPHOS such as atovaquone (AQ), a well-tolerated anti-malarial drug that is effective for prevention/treatment of *pneumocystis jiroveci pneumonia* (PJP) [16]. Prophylaxis for PJP is standard in pediatric oncology and strongly recommended amongst adult AML patients [17,18]. Our data align with published literature suggesting that AQ inhibits electron transport chain complex III (ETCIII), which leads to OXPHOS suppression and unregulated activation of the integrated stress response [16,19,20,21]. Using AQ to disrupt this dependency may prevent relapse and improve the efficacy of existing therapeutics in children, without additional myelotoxicity.

Our previous data show that AQ has impressive anti-leukemia effects. Specifically, we have shown that AQ induces apoptosis of AML cells in vitro and prolongs survival when administered as a single agent in patient-derived xenograft (PDX) mouse models harboring AML [16]. AQ is particularly effective in in vitro experimental conditions that mimic the bone marrow microenvironment, which is remarkable as these conditions typically reduce the efficacy of conventional and targeted treatment agents [22,23]. The side effect profile for AQ is desirable as it does not cause myelosuppression even in hematopoietic stem cell transplant patients and is safe for patients with hepatic or renal dysfunction [24]. Xiang et al. reported reduced relapse rates in adults with AML who received AQ for PJP prophylaxis instead of other agents [25].

The strong pre-clinical data, the favorable side effect profile, and its approval in the US by the Food and Drug Association (FDA) for PJP prophylaxis, makes incorporation of AQ into therapy for pediatric AML particularly attractive. However, efficacy of AQ against PJP is dependent upon achieving sufficient plasma concentrations [26,27,28] Because AQ is administered enterally and intensive cytotoxic chemotherapy can lead to oral aversions and poor absorption through the gut mucosa, our primary aim was to evaluate the feasibility of incorporating AQ into intensive upfront treatment of newly diagnosed pediatric AML patients. Here, we show that AQ is well tolerated by children and administration does not cause an excessive burden on families. We also show that when standard dosing is followed, target plasma concentrations of AQ, during intensive chemotherapy, are frequently not achieved. To address this issue, we now have a laboratory developed test available from our Clinical Laboratory Improvement Amendment of 1988 (CLIA)-certified laboratory to provide AQ plasma concentration data [29]. Lastly, correlative biology studies on patient samples from the trial provided detailed information on in vitro responses to AQ, on measures of effects on OXPHOS, mTOR and STAT3, and in summation provide a strong rationale for further evaluation of AQ and other medications that inhibit OXPHOS for pediatric AML.

## 2. Materials and Methods 

### 2.1. Trial Design

Patients with de novo AML were enrolled at Texas Children’s Hospital (Houston, TX, USA) and Johns Hopkins Children’s Center (Baltimore, MD, USA). In this single-arm trial, patients received standard-of-care United Kingdom Medical Research Council (MRC)-based Induction chemotherapy. Some modifications to therapy were allowed; however, inclusion of cytarabine and daunorubicin (DA) at particular dosing was required (Appendix A). Daily AQ was administered during standard induction chemotherapy for AML (Figure 1A). Additionally, all patients with FLT3-ITD received targeted therapy with sorafenib starting on day 11, and all patients received standard of care supportive measures with prophylactic antibiotics, antifungals, and antiemetics. Daily AQ began on day 6 and dosing was based on established PJP prophylaxis per Hughes et al. as follows: ages 1–2 months 30 mg/kg/dose, ages 3–24 months 45 mg/kg/dose, ages >24 months to 13 years 30 mg/kg/dose with a maximum of 1500 mg daily, and 1500 mg daily for children >50 kg or over 13 years of age [26]. Doses vomited within 30 min of administration were repeated. AQ was obtained from a clinical commercial supply. All patients had diagnostic FISH, karyotype, and clinical NGS. Feasibility of AQ incorporation was assessed through adverse event (AE) reporting (per NCI CTCAE v5 [www.ctep.cancer/gov]), and daily parent/caregiver ease of administration scores (scale: 1–5, 1 = very difficult, 5 = very easy to administer, Appendix A). Disease response assessment was performed via bone marrow samples collected at the end of Induction 1 (EOI) and analyzed via local-institutional standard morphology, flow cytometry (for minimal residual disease (MRD)) and FISH. A complete response (CR) was defined as <5% blasts in the bone marrow by flow cytometry with no peripheral blood evidence of disease. All gastrointestinal (GI) AEs ≥ grade 2 were collected, in addition to other AEs ≥ grade 4. Patients who were administered at least 85% of planned doses and missed fewer than 2 consecutive doses of AQ were eligible for plasma concentration/PK analyses. The expectation was that 95% of patients who received at least 85% of planned doses of AQ would have plasma concentrations >40 µM by day 22. Based on this expectation a sample size of 15 provided 80% power to estimate a success rate of greater than 75% with the assumption of an α error 10% and a 95% confidence interval [30]. Sample size was increased beyond 22 to allow for enhanced descriptive statistical analysis of plasma concentrations of AQ at sequential time points. Correlative biology studies assessed AQ-induced apoptosis at 30 µM, AQ effects on OXPHOS, and AQ effects on relevant signaling activity. Additionally, patient-derived xenografts (PDX) were established and treated with AQ.

### 2.2. Patients

Eligible patients included those ≥1 month and <21 years of age with de novo AML per the 2008 WHO Myeloid Neoplasm Classification, myeloid sarcoma or treatment-related AML who had not yet been treated for their myeloid neoplasm [31]. Patients were required to have adequate liver function and to be able to tolerate enteral medications in the opinion of the treating physician at the time of enrollment. Detailed eligibility criteria are provided in the Appendix A. Baylor College of Medicine (Houston, TX, USA) served as the coordinating center for this trial which is registered at www.clinicaltrials.gov (NCT03568994, accessed on 19 February 2023) and approved by the local institutional review boards (BCM IRB# H-42961) at all participating centers. Written informed consent (and assent as appropriate) was obtained for treatment and for optional correlative biology studies from patients and/or from parents/legal guardians according to institutional, local and federal policies and in accordance with the Declaration of Helsinki. 

### 2.3. Adverse Events 

For the purpose of correlation with other parameters, AEs were evaluated based on the daily AE burden (sum of all AE grades for all AE types given in each day of Induction cycle) and total AE burdens (TB). The TB value accounts for AE grade and duration (TB = daily AE burden/number of Induction days (30 days)) [32]. Adverse events were categorized into 9 types ((1) nausea, (2) vomiting, (3) diarrhea, (4) metabolic and nutritional disorder—mainly anorexia, (5) mucositis, (6) colitis, (7) stomach pain, (8) other gastrointestinal disorders, and (9) all other events). The TBs for each patient based on these AE categories can be found in Appendix A. 

### 2.4. Atovaquone Plasma Concentration Measurement

Peripheral blood for AQ plasma concentration measurements were drawn during Induction 1 prior to starting chemotherapy on Days 0 and 1 and then prior to the administration of AQ on Days 6, 11, 13, 15, 18, 20, 22 and 29. Bone marrow for AQ concentrations was collected with the EOI bone marrow evaluation for response assessment. Electrospray ionization-mass spectrometry and HPLC retention time confirmed the presence of intact AQ in patient samples. Calculation of the concentration of AQ in each sample was performed by creating a standard curve by adding fixed concentrations of AQ to no-drug control samples. Concentrations of AQ in patient samples were calculated using relative HPLC peak areas. Appropriate positive and negative controls were included for each batch of samples run. Concentrations >10 μM (3.7 mcg/mL) were considered to meet the anti-leukemia threshold based on our prior work and concentrations >40 µM (15 mcg/mL) were considered to meet the PJP prevention threshold [16,27]. Patients were evaluable for assessment of time to steady state concentration of AQ if they had plasma collected for testing on days 15, 22, and 29.

### 2.5. Cell Lines

Cell lines were obtained from ATCC (Manassas, VA, USA) or colleagues. The human bone marrow stromal cell line, HS5, was used to mimic the supportive conditions provided by soluble factors within the bone marrow microenvironment [23]. All cell lines used were validated by short tandem repeat authentication at least annually. 

### 2.6. Reagents

Atovaquone was purchased from Sigma (A7986) for in vitro work and purchased from Amneal Pharmaceuticals for in vivo studies. Fluorochrome-conjugated antibody to pS6-AF647 (Ser240/244) was from Cell Signaling Technologies, CD130-AlexaFluor 488 was from Abcam, pY-STAT3-PE and isotype control were from BD Biosciences. 

### 2.7. Annexin V and Phosphoflow Assays

Samples from enrolled patients were incubated with PBS (vehicle) or AQ (30 µM) for 24–96 h. Primary samples with baseline viability >60% were treated with 30 µM AQ dissolved in medium, or vehicle control, for 24–96 h. We measured AQ-induced apoptosis of fresh AML samples from most enrolled patients (Appendix A). Primary samples were supported in vitro with HS5 conditioned media (CM), co-cultured with HS5 cells separated by a polyester transwell membrane with a 0.4 µm pore size (Corning, Corning, NY, USA, Fisher Scientific, Hampton, NH, USA), or cultured alone in standard Iscove’s modified Dulbecco medium (IMDM) with 20% FBS. Spontaneous and AQ-induced apoptosis were quantified by Annexin V-FITC and propidium iodide positivity, and MV4-11 cells were used for positive controls. Phosphoflow was conducted as previously described with Kasumi-1 cells as the positive control for CM-induced signaling [33].

### 2.8. The Oxygen Consumption Rate

The oxygen consumption rate (OCR) was measured using the Seahorse XF96 analyzer (Seahorse Bioscience, North Billerica, MA, USA). Study patients AML bone marrow cells were incubated with vehicle or increasing doses of AQ for 3 h at a 1:1 ratio of IMDM and CM. Plates were coated with Cell-Tak (Fisher Scientific), and then AML or normal bone marrow cells were seeded after being suspended in Seahorse medium. The OCR was measured sequentially before and after injection of (1) oligomycin (1 µM), (2) carbonyl cyanide 4-(trifluoromethoxy) phenylhydrazone (FCCP) (1 µM), and (3) antimycin A (1 µM) and rotenone (1 µM) (Sigma-Aldrich, St. Louis, MI, USA).

### 2.9. Xenograft Model

Eighteen 8-week-old NOD-scid IL2Rgnull-3/GM/SF (NSGS) mice (female, *n* = 10; male, *n* = 8) were injected by tail vein with 2 × 10^5^ primary AML cells from patient Unique Patient Number (UPN) ATACC 10. Treatment with AQ (200 mg/kg per day; female, *n* = 5; male, *n* = 4) or VC (vehicle control; female, *n* = 5; male, *n* = 4) by daily oral gavage begun on the same day as cell injection given the long time to achieve steady-state drug concentrations. The mice were randomized into treatment and control groups. Peripheral blood human CD45-APC-H7 and CD33-AF700 percent positivity by flow was used to quantify disease burden every 1 to 2 weeks once signs of engraftment were observed. The mice were followed for body condition and survival as previously described [16].

### 2.10. Statistical Analysis

Statistical analysis was performed using GraphPad Prism software (version 9.1.1; GraphPad Software, La Jolla, CA, USA), R (version 3.6.3, R Core Team, 2021). Pearson correlation was used to compare plasma concentrations, ease of administration scores, AE burden and weight loss. Heatmaps were made using Heatmap.2 of the gplots package available in R [34]. In order to account for age-based variation in plasma concentrations and ease of administration scores, patients were divided into 3 age groups of approximately similar size (0 < age ≤ 3 years (6 patients), 3 < age ≤ 15 years (10 patients), and 15 < age ≤ 21 years (8 patients)) (Figure 1B). The significance of observed differences between atovaquone and vehicle treated samples in vitro and in vivo was determined by repeated measures ANOVA to compare all samples at all tested time points. For in vivo survival analyses, Kaplan–Meier survival curves were generated and compared for significant differences by the log-rank test.

## 3. Results 

### 3.1. Patient Characteristics

Twenty-six patients were enrolled between 23 July 2018 and 2 September 2020 (Table 1, Appendix A). A total of 8 of 26 patients self-identified as White-Hispanic, 4 of 26 as Black, and 1 as Asian. The remaining 13 patients self-identified as non-Hispanic White. A total of 12 were classified as being at higher risk of relapse per the currently enrolling Children’s Oncology Group study for de novo AML AAML1831 (NCT04293562) definitions (Appendix A). The patient cohort included four children with non-Down Syndrome acute megakaryoblastic leukemia (AMKL) and one with myeloid sarcoma (UPN ATACC 15). Most patients (*n* = 15) were CNS positive at diagnosis. A total of 5 of 26 patients had core binding factor fusions, 5 had *KMT2A*-rearrangements, 5 harbored IDH mutations, and 6 had FLT3-ITD with allelic ratio > 0.1. A total of 8 of 26 patients had MRD at the EOI > 0.1%. Twenty-five patients received cytarabine, daunorubicin and etoposide (ADE) + dexrazoxane for cardioprotection, and one patient received DA + gemtuzumab ozogamicin (GO) + dexrazoxane (Figure 1A, Appendix A). Though our trial was not powered to evaluate clinical outcomes, of the 26 patients, 13 have had events (11 with relapses or induction failures and two patients suffering treatment-related mortality as their first event due to complications related to hematopoietic stem cell transplant (HSCT)). The remaining 13 patients have remained in first complete remission (CR1) and have a follow-up time of >26 months (median: 28 months). Of the patients in CR1, 3 underwent HSCT in their first CR. In total, 20 of 26 are alive at a median follow up of 30 months. Of the three patients with induction failure, all subsequently achieved MRD negative status and proceeded to HSCT. One remains in CR1 and two relapsed after transplant, one of whom remains in second CR. The event-free survival (EFS) and overall survival (OS) outcomes seen among these patients are consistent with those from recent COG trials [3,4].

### 3.2. Atovaquone Plasma Concentrations Varied among Pediatric AML Patients

A total of 24 out of 26 enrolled patients were eligible for inclusion in the evaluation of AQ ease of administration and determination of AQ plasma concentrations. Two patients (UPN ATACC 2 and UPN ATACC 11) were excluded due to medical complications that arose during the first 5 days of Induction preventing administration of all enteral medications. Plasma concentrations of AQ varied widely but increased throughout the cycle in the majority of patients. The majority (83%, 20/24) of evaluable patients achieved target anti-leukemia AQ plasma concentrations (>10 µM) (Figure 1C,D). However, AQ plasma concentrations did not achieve historically defined targets for PJP prophylaxis/treatment (>40 µM) in the majority of patients (Figure 1C). Seven patients (29%) achieved a plasma concentration above the target threshold for anti-PJP effect during Induction with the median time to achieving this target of 23 days, with only 1 patient achieving these concentrations consistently (UPN ATACC 8, Figure 1C,D). Plasma concentrations in bone marrow samples collected at the EOI reflected results similar to that of the peripheral blood (Appendix A, Figure 1D). When evaluating AQ concentrations by age group, we found that the average peripheral blood plasma concentrations trended upward toward the EOI in all age groups (Figure 1E). The range of concentrations achieved showed higher variability in the oldest group of patients compared to the younger age groups as shown in Figure 1D. However, at each sampling time point the three age groups did not show statistical significance when compared to each other by Student’s *t*-test. 

### 3.3. Ease of Atovaquone Administration Did Not Correlate with Plasma Concentrations Achieved

Next, we wanted to know if issues with administration contributed to low AQ plasma concentrations among our patients. Ranking of the ease of administration of AQ were evaluated via diaries completed by the patient or caregiver. Seventy-one percent (17/24) of the evaluable patients had average ease of administration scores >3 (more easy than difficult) out of 5. Toward the end of the cycle, the majority of the patients provided scores of 5 (very easy) (Figure 2A). Overall, patients in the oldest age group (15 to 21 years) provided the greatest ease of administration according to self-assessment. The youngest age group of patients experienced some difficulty initially with scores of 2.9 in the first 5 days of administration (cycle days 6–11), but all patients in the group indicated easier administration by the EOI (Figure 2A). In comparison the average 5-day post-administration score for grade school-aged children was 3.6 and 3.9 for the young adult patients. Additionally, the average ease of administration scores for each age group were all >3, indicating that patients and families did not find administration of AQ to be particularly burdensome (Figure 2B). We did not observe significant correlations between the ease of administration of AQ and the plasma concentrations detected in our patients. 

### 3.4. Adverse Events Did Not Correlate with Atovaquone Plasma Concentrations

To further evaluate the feasibility of incorporating AQ within standard pediatric AML treatment, we analyzed AEs that occurred for each patient during the Induction cycle. Gastrointestinal AEs have the potential to directly influence the absorption of AQ. Therefore, we collected data on all ≥ grade 2 GI-related AEs. Some of the common chemotherapy-related gastrointestinal AEs such as diarrhea, nausea and colitis were the most reported events during the Induction treatment cycle (Figure 3A). Among all the GI related AEs reported, 70% of them were grade 2 events (Figure 3B), but none of the events were classified as likely or probably related to the use of AQ. We found no correlation between AEs and the ease of administration scores (Appendix A). To better evaluate AEs, the types of events and number of events that occurred on the same day were compiled to generate a daily “adverse event burden”. The adverse event burden for each patient varied throughout the cycle with no clear pattern (Figure 3C) and did not correlate with plasma concentrations or ease of administration scores (Appendix A). 

### 3.5. AQ Induced Apoptosis in the Majority of Patient Samples Tested Ex Vivo

A total of 20 of 26 patients had sufficient diagnostic peripheral blood, bone marrow, or pheresis material to allow for correlative biology testing. Based on our prior work demonstrating that AQ induces more apoptosis in conditions that mimic the bone marrow microenvironment, patient samples were tested for in vitro AQ-induced apoptosis in three separate conditions, with standard media + 20% FBS, with media that had been conditioned by HS5 cells, and with co-culture with HS5 cells across a transwell membrane [16]. Consistent with our previously published work, we found that AQ induced minimal apoptosis without stroma-derived soluble factors (Figure 4A) [16]. Conditioned media and transwell co-culture support decreased spontaneous apoptosis and enhanced AQ-induced apoptosis in aggregated sample data (Figure 4A) and in individual patient samples (Figure 4B, Appendix A). Additionally, for two patient (UPN ATACC 5 and 6) samples were evaluated for AQ-induced apoptosis with both fresh sample and with sample that had been viably frozen immediately after collection and then thawed prior to testing (Appendix A). Spontaneous apoptosis was much higher in samples that had been viably frozen, making discernment of the amount of apoptosis attributed to AQ challenging. Non-responders, defined as having less than 15% AQ-induced apoptosis, were rare. In total, 16 of 20 patient samples demonstrated in vitro AQ-induced apoptosis ≥15% and were therefore classified as responders (Appendix A).

### 3.6. Single-Agent AQ Delayed Disease Progression and Prolonged Mice Survival In Vivo

Patient-derived xenograft models were developed for several of the patients who had samples banked. The UPN ATACC 10 PDX model was used to evaluate the single-agent efficacy of AQ. The treatment schema is depicted in Figure 4C. Due to the prolonged time required to achieve steady state plasma concentrations of AQ, dosing was started on the same day as tail vein injection of AML cells. Consistent with our prior published work [16], single-agent AQ was well tolerated with no weight loss of mice until the late stages of disease progression (Appendix A), and female mice demonstrated earlier engraftment compared to males (Appendix A). Mice treated with AQ demonstrated significant delays in disease progression as shown in representative flow plots of VC- and AQ-treated male mice from the same litter that were cohoused. Composite flow cytometric data for human CD45 and CD33 demonstrated decreased peripheral blood disease in AQ-treated mice (*p* < 0.0001 by ANOVA, Figure 4D,E). Mice treated with AQ had significantly prolonged survival compared to mice treated with VC, with median survival of AQ-treated mice of 91 days vs. 69 days (*p* < 0.05 by log-rank test, Figure 4F, Appendix A).

### 3.7. AQ Dramatically Suppressed Oxygen Consumption in Primary AML Blasts 

Our prior evaluations suggested that the mechanism of AQ-induced apoptosis is based on suppression of oxidative phosphorylation, so the effect of AQ on the OCR was tested in patients with available samples (*n* = 17). AQ exposure at 3 increasing doses for 3 h prior to evaluation of the OCR demonstrated significant dose-dependent reductions in the OCR for all patients as shown in composite and individual datasets (Figure 5A,B and Appendix A). AQ did not notably change extracellular acidification rate. Several patients had a minimal measurable basal OCR, possibly due to lower blast numbers in the sample or minimal metabolic activity of the bulk leukemia cell population outside of the host microenvironment. Visual inspection of the effect of AQ on the OCR in comparison to the ex vivo AQ-induced apoptosis response assessment appeared to show a higher baseline OCR and greater AQ-induced OCR suppression in patient samples that had been classified as responders. 

### 3.8. Phosphoflow Demonstrated Minimal Effects of AQ on pY-STAT3 and p-S6

AQ has been reported to reduce IL-6-induced STAT3 activation (pY-STAT3) by downregulating the IL-6 receptor transmembrane subunit, gp130 [25]. Therefore, we conducted surface and phosphoflow analyses of patient samples to evaluate whether AQ inhibited soluble factor-stimulated pY-STAT3 and p-S6 activity, as well as to determine if AQ decreased surface expression of gp130. Across all samples there was minimal suppression of p-S6 and pY-STAT3 by AQ (30 µM × 3 h) when administered prior to stimulation with media conditioned by HS-5 stromal cells. AQ did not alter expression of gp130, indicating that downregulation of gp130 is not the primary mechanism of action of AQ in these patient samples. Representative data from UPN ATACC 10 are shown in (**C**–**E**), and all signaling data are shown in Appendix A.

### 3.9. Evaluation of Characteristics of Responders vs. Non-Responders

In the ATACC AML correlative biology cohort, 16/20 patient samples demonstrated >15% ex vivo AQ-induced apoptosis (Figure 5F). Because of the rarity of non-responders, we sought to evaluate for potential associated characteristics to better understand which cases are unlikely to show AQ-induced apoptosis. Though the numbers of samples were small, samples with >15% AQ-induced apoptosis (*n* = 15) had a trend towards a higher OCR when compared to samples with <15% AQ-induced apoptosis (*n* = 2) (Appendix A and Figure 5G). Additionally, we evaluated our cohort for potential associations between AQ-induced apoptosis and known cytogenetic features and mutations within AML blasts (Figure 5H, Appendix A). All patients with FLT3-ITD (*n* = 3), *KMT2A*-rearranged (*n* = 6), and IDH mutations (*n* = 3) were classified as having a leukemia response to AQ. Interestingly, the leukemic blasts of all 3 patients with t(8;21) were classified as not responding to AQ. The other patient whose leukemia blasts did not respond to AQ had an NPM1 mutation. Our prior pre-clinical work with AQ also demonstrated enrichment of *KMT2A*-rearranged and FLT3-ITD leukemias amongst AQ-responsive patient leukemia samples [16].

## 4. Discussion

Our data demonstrate the feasibility of combining AQ with traditional chemotherapy for pediatric AML. Patients of all ages were able to tolerate AQ and no AEs were attributable to AQ administration. Importantly, while ease of administration scores were suboptimal initially, especially in the baby/toddler age group, by the end of Induction 1 we found that ease of administration improved significantly. The fact that 23 of 24 patients who received AQ opted to continue AQ after the study period had completed is important additional supportive evidence of the tolerability of AQ in the pediatric population, even though the study protocol did not mandate its continuation after Induction 1. These data support the feasibility of administering AQ during intensive upfront therapy for pediatric AML without placing an undue burden on patients and families.

Amongst study patients, AQ plasma concentrations in the range of in vitro anti-leukemia efficacy (>10 µM) were frequently achieved, but concentrations of >40 µM required for PJP prophylaxis at standard dosing were rare during Induction 1. In considering the data, the finding that our patients did not achieve suggested anti-PJP concentrations should not have been surprising. In addition to the fact that atovaquone has poor bioavailability, initial dosing suggestions for children with hematologic malignancies are exclusively based on studies performed on children with HIV [24]. Differences in the treatment of patients with HIV and hematologic malignancies, such as concomitant medications and comorbidities, could play an unknown, yet significant, role in drug disposition. While there are studies which show that atovaquone, at this dosing, is a reasonable alternative to trimethoprim-sulfamethoxazole in preventing PJP in children with cancer, they are small and only consider clinical outcomes [35]. This idea of our population not being able to achieve desired concentrations at currently suggested dosing is supported not only by our results, but in one of the few studies identified studying atovaquone PK in patients which included those with hematologic malignancies, most patients also did not achieve desired anti-PJP concentrations of atovaquone at steady state. Robin et al. investigated atovaquone concentrations at steady state in 33 adult patients, 79% of which had hematologic malignancies or were hematopoietic transplant recipients and found that at standard dosing, 58% percent of patients did not achieve desired atovaquone concentrations [36].

Surprisingly, low plasma concentrations of AQ did not correlate with the presence of GI-related AEs or weight loss and so our data do not support that the absence of toxicity is predictive of target concentrations of AQ. Rather, if AQ is being used for both anti-leukemia effect and for PJP prevention in patients receiving intensive chemotherapy, following AQ plasma concentrations is highly recommended. In order to facilitate testing of AQ concentrations and to fill a gap in currently available tests, our institution has developed a mass spectrometry-based assay for measuring plasma concentrations of AQ [29]. The availability of this assay with a 24 h turnaround time will allow for evaluation of the impact of dose adjustment of AQ when given during intensive chemotherapy regimens to achieve target plasma concentrations (trial in development). For other antimicrobials, such as TMP-SMX, an appropriate prophylaxis level is lower than what is needed to treat active infection. Therefore, it’s possible that lower levels of AQ are sufficient to prevent PJP. However, no studies have been conducted to answer this question and given the rarity of PJP this is likely a challenging study to conduct. While there are not available data on the target plasma concentrations of AQ needed to achieve adequate prophylaxis of PJP, there have been several studies supporting the need to achieve therapeutic concentrations for adequate PJP prophylaxis. The initial therapeutic dose finding studies published in 1993 found that zero of six patients had resolution of PJP when AQ plasma concentrations were in the 0–10 µM range. These poor outcomes are compared to 95% of patients with therapeutic efficacy at an AQ plasma concentration of at least 41 µM [27]. Several studies have demonstrated that AQ at standard therapeutic dosing is an effective agent for prevention of PJP [35,37,38,39,40,41]. However, these studies did not quantify the AQ plasma concentrations of patients. Furthermore, subjects in the majority of these studies were not typically receiving intensive chemotherapy. In contrast, low dose AQ administered at either 50% dosing or thrice weekly has demonstrated insufficient efficacy as prophylaxis, suggesting that low plasma concentrations of AQ may put patients at risk for breakthrough PJP infections [42]. Given the severity of PJP infection in children with AML, increasing intensity of contemporary AML treatment protocols, and data from our trial demonstrating that 71% of patients did not achieve levels of ≥41 µM with standard AQ dosing, we advise monitoring AQ plasma concentrations to ensure they are within the therapeutic range that is known to provide effective prophylaxis. As providing PJP prophylaxis in pediatrics is standard, we also suggest consideration of dual coverage for PJP prophylaxis until sufficient levels of AQ have been established. Further studies are planned to use plasma concentrations of AQ to adjust dosing, as is performed frequently with antifungals to better quantify AQ dose ranges in pediatric AML patients during and after intensive chemotherapy and to evaluate a dose adjustment algorithm to achieve target levels for PJP prophylaxis during intensive chemotherapy cycles.

Our correlative biology results support suppression of OXPHOS as the primary mechanism of action by which AQ exerts its anti-leukemia effect. Furthermore, our correlative data support the potential benefit of inclusion of AQ or other drugs that inhibit OXPHOS into pediatric AML treatment and highlights the importance of the BM microenvironment in assessing responses to leukemia-directed therapy. A randomized controlled trial to definitively demonstrate an improvement in survival for patients receiving AQ is impractical in pediatric AML given the rarity of the disease. However, data regarding current PJP prophylaxis practices are being collected through the Children’s Oncology Group Phase III trial for de novo AML (NCT04293562, PI: Cooper), which may allow comparisons of outcomes between patients receiving different prophylaxis regimens as was reported for adult BMT patients [25]. Further preclinical studies are needed to determine the in vitro and in vivo efficacy of AQ in combination with cytotoxic chemotherapy and/or venetoclax, to evaluate the effect of AQ on leukemia stem cells in vivo, to explore the best timing of administration of AQ, and to better identify which patients are likely to derive the most benefit from AQ or other agents targeting OXPHOS. 

## 5. Conclusions 

All children with de novo acute myeloid leukemia (AML) receiving standard-of-care therapy require prophylaxis against *pneumocystis jiroveci pneumonia* (PJP), which is usually given in the form of trimethoprim/sulfamethoxazole or pentamidine, but atovaquone is also FDA approved for this purpose. Atovaquone’s major drawbacks include palatability and need for daily administration. However, there is renewed interest in atovaquone as an inhibitor of mitochondrial OXPHOS, a metabolic pathway on which chemoresistant AML cells are dependent. Atovaquone induces AML cell apoptosis and prolongs survival in pre-clinical AML models. We show, in a clinical pilot study, that combining atovaquone with standard AML induction chemotherapy is feasible and safe in children. Using PJP-based dosing generally achieves adequate concentrations necessary for its anti-leukemic effect. Administration of atovaquone during AML therapy may achieve dual effects, providing PJP prophylaxis and an anti-leukemic effect, providing an advantage over other methods of PJP prophylaxis.

## Figures and Tables

**Figure 1 cancers-15-01344-f001:**
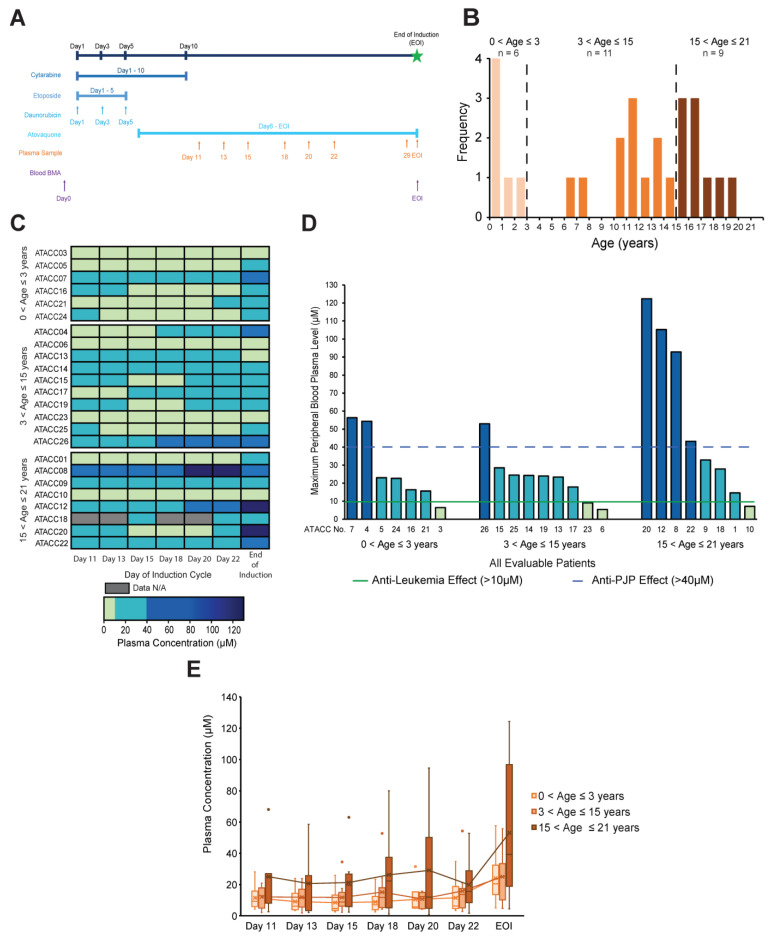
**Plasma concentrations of atovaquone during treatment.** (**A**) Schema of Induction 1 treatment period including administration of chemotherapy and sampling times. (**B**) Age distribution of patients included in the clinical trial. (**C**) Overview of plasma concentrations (μM) in the peripheral blood for each of the 24 patients (grouped by age range based on (**B**) after starting atovaquone administration. Concentrations were measured on days 11, 13, 15, 18, 20, 22, and at the end of induction (EOI). (**D**) Maximum peripheral blood plasma concentrations (μM) achieved for each patient. (**E**) Whisker plot showing the ranges and averages atovaquone peripheral blood plasma concentrations achieved within each age group. X indicates mean concentration; top and bottom vertical lines indicate maximum and minimum values, respectively; top horizontal line of the box indicates 3rd quartile, bottom horizontal line indicates 1st quartile.

**Figure 2 cancers-15-01344-f002:**
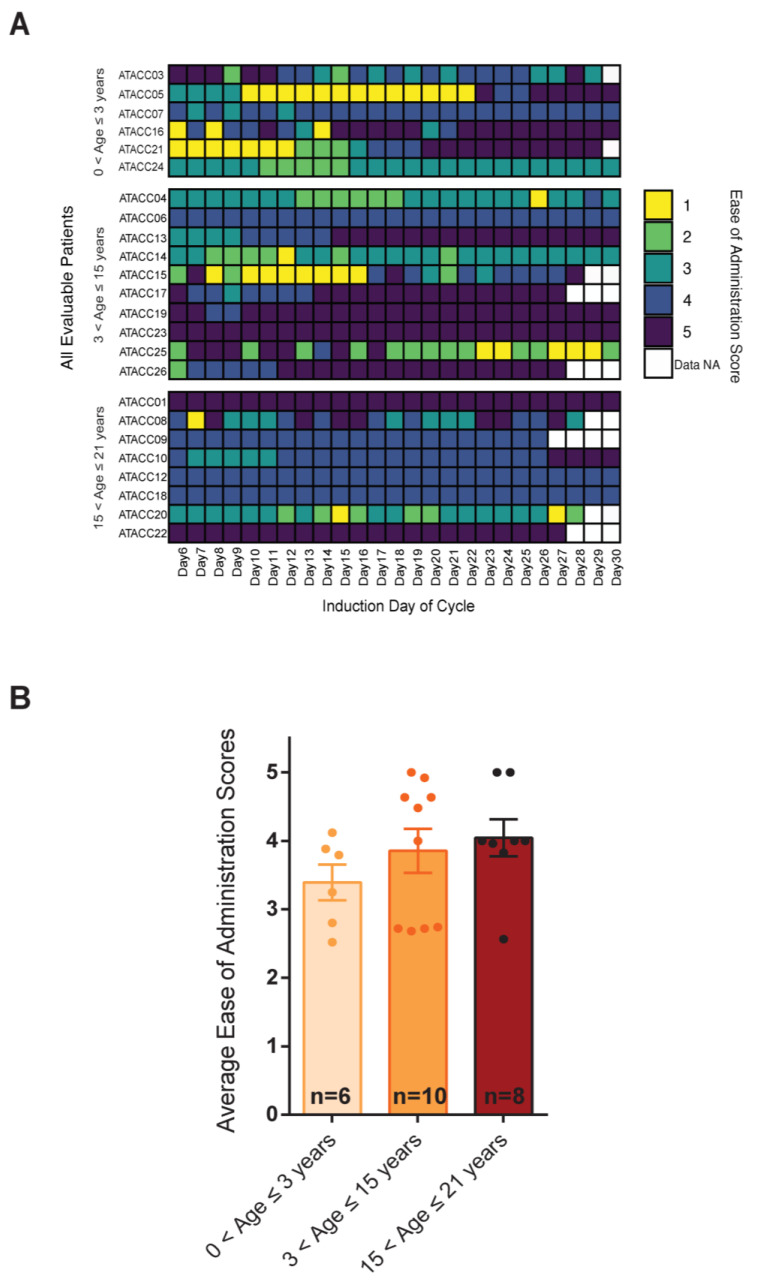
**Distribution of ease of administration scores.** (**A**) Overview of ease of administration score distribution during administration of atovaquone for all evaluable patients (24) on study, grouped by their age groups. Scoring scale: 1 = very difficult to administer (yellow) to 5 = very easy to administer atovaquone (dark blue). (**B**) Average ease of administration scores for each age group of patients (Error bars represent SEM).

**Figure 3 cancers-15-01344-f003:**
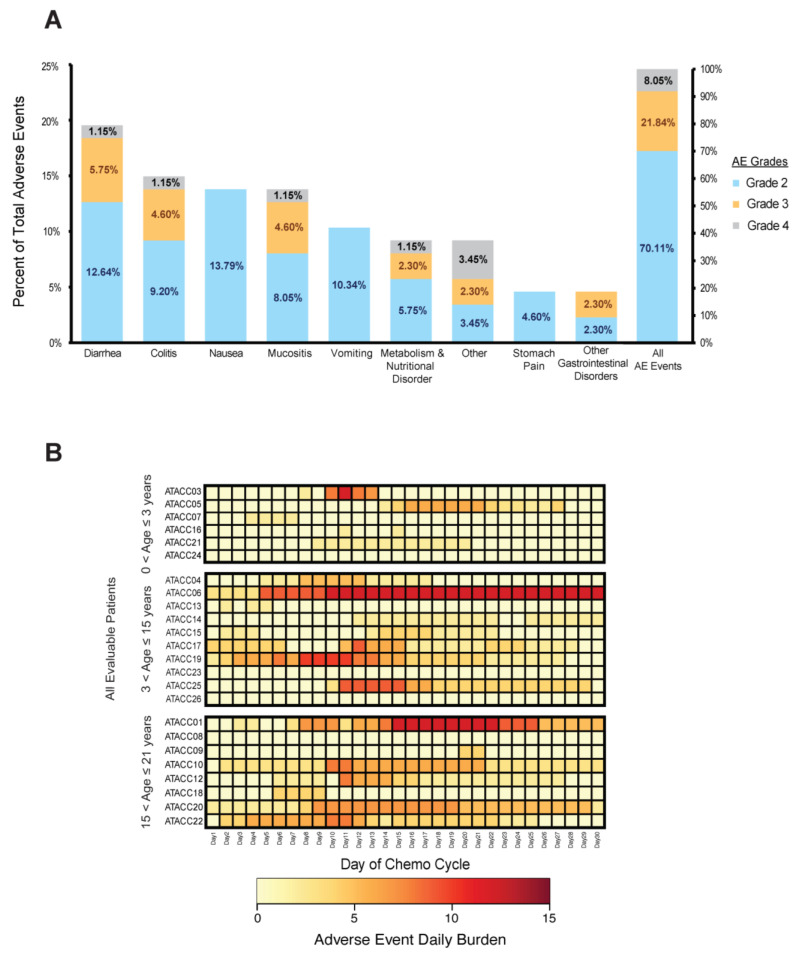
**Gastrointestinal toxicity adverse events during atovaquone treatment.** (**A**) Distribution of all GI-related adverse events reported on the trial. Each bar represents the percentage that each AE type was observed during the trial; small boxes represent the grades that were recorded for each AE type; percentages in the box represents the chance of an AE at a specific grade occurred during this trial. (**B**) Heatmap of adverse event daily burden (sum of all AE grades assigned each day) for each evaluable patients (*n* = 24) throughout the treatment period, patients are group into their respective age ranges.

**Figure 4 cancers-15-01344-f004:**
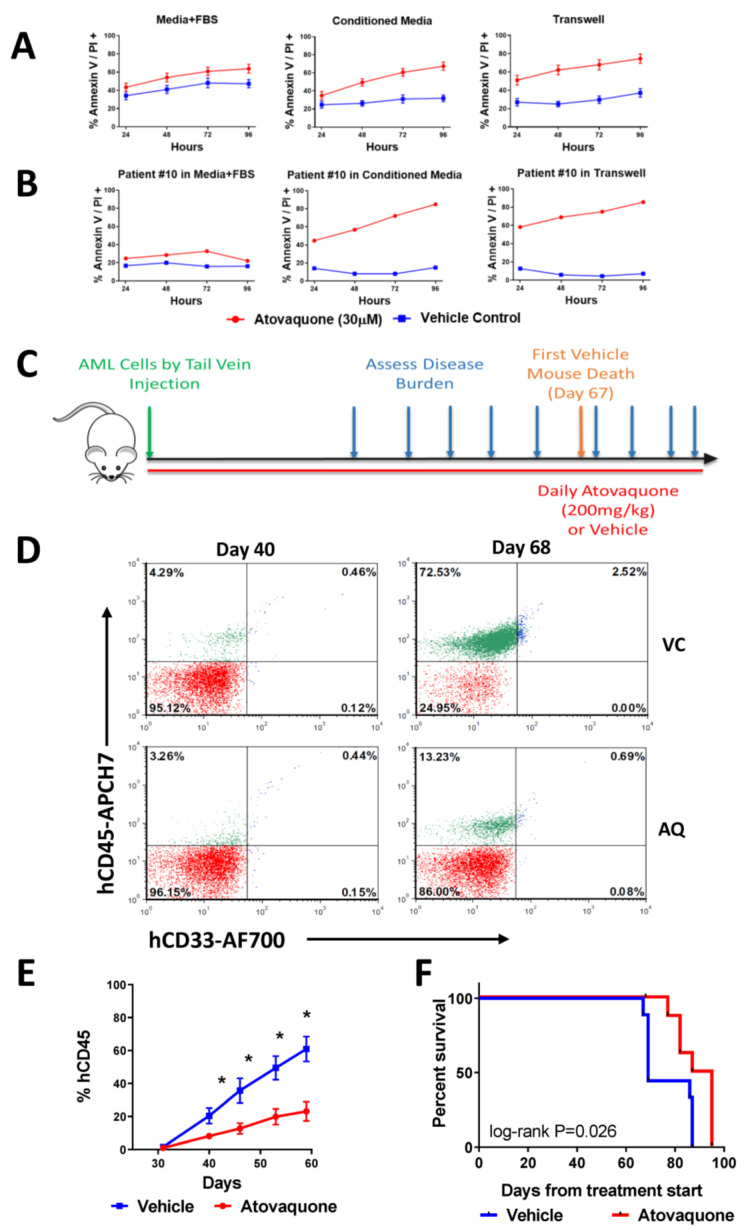
**AQ induces apoptosis for ex vivo patient samples and Single-agent treatment with AQ decreases disease burden and prolongs survival in a patient-derived xenograft model.** (**A**) Composite data on spontaneous vs. atovaquone induced apoptosis of primary patient samples incubated in different conditions: media with 20% fetal bovine serum (FBS), HS-5 conditioned media, or co-cultured with HS-5 stromal cells separated by a transwell mesh insert. Apoptosis was quantified by flow cytometry using annexin V-fluorescein isothiocyanate (FITC) and propidium iodide (PI). (**B**) Representative data from Patient 10 in different conditions shown. Data from Patient 10 was generated using a fresh sample. (**C**) NSGS mice were injected by tail-vein with 2 × 105 cell from the diagnostic specimen from patient 10. Treatment with atovaquone (AQ) (200 mg/kg per day; *n* = 9[(female, *n* = 5; male, *n* = 4]) or vehicle (VC) (*n* = 9 [female, *n* = 5; male *n* = 4]) by daily oral gavage began on the day of cell injection. Disease burden was assessed by flow cytometric data throughout treatment. (**D**) Representative dot plots of human CD45 and CD33 at 2 time points (days 40 and 68) are shown for 1 vehicle-treated and 1 atovaquone-treated mouse. Depicted mice were males from the same litter and were cohoused. (**E**) Composite flow cytometric data for human CD45 and CD33 for all treated mice demonstrated decreased peripheral blood disease in atovaquone treated mice. Bars represent the means and standard errors of the mean, * *p* < 0.05 by Student *t* test (*p* < 0.0001 by analysis of variance). (**F**) Survival analysis of the mice demonstrated improved survival in atovaquone-treated mice compared with vehicle. *p* < 0.05 by log-rank.

**Figure 5 cancers-15-01344-f005:**
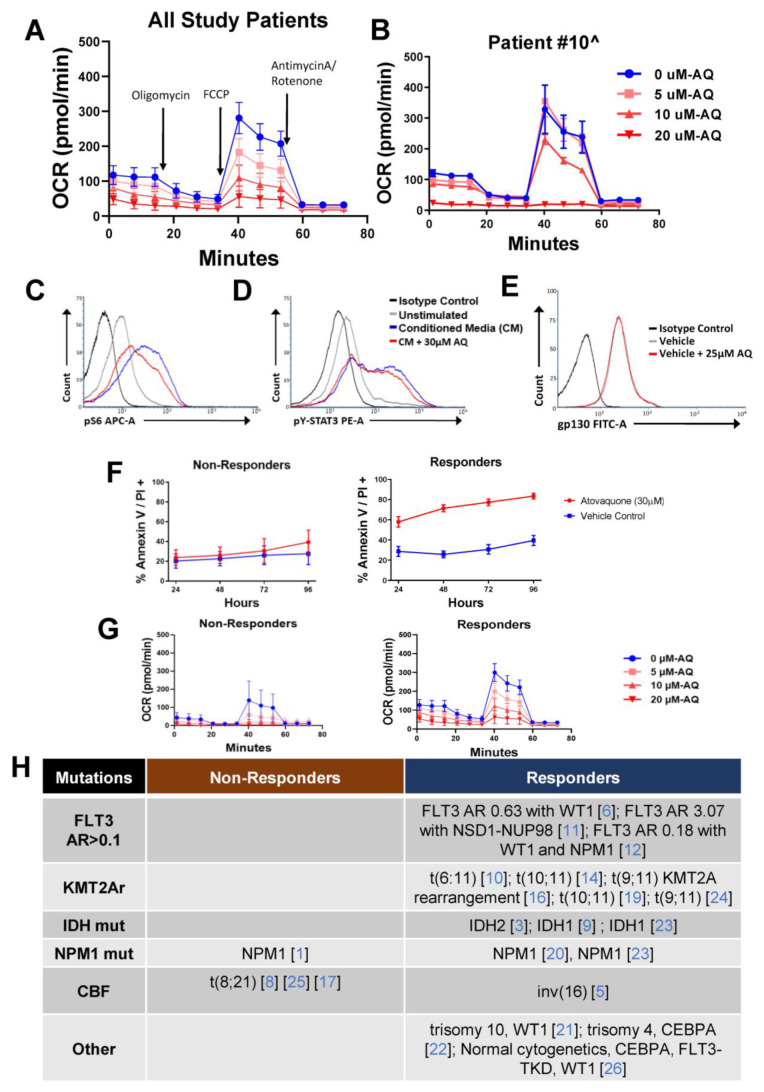
**Atovaquone induces a dose-dependent suppression of basal and maximal mitochondrial respiration of AML primary samples, has minimal effect on pS6 and pYSTAT3, and evaluation of the OCR and cytogenetics/mutations for non responders vs. responders to ex vivo AQ.** (**A**) Dose-dependent reductions in the oxygen consumption rate (OCR) of primary AML cells after 3 h treatment with increasing doses of atovaquone (AQ). Doses of AQ evaluated were 0, 5, 10, and 20 μM and composite data from all study patients (*n* = 17) shown. The OCR for vehicle treated vs. all doses of AQ tested were statistically different (*p* < 0.001 by ANOVA). (**B**) Dose-dependent reductions in the OCR for patient 10 as a representative patient sample. Patient samples were rested in HS5 conditioned medium for 24 h prior to 3 h incubation with indicated doses of AQ. (**C**,**D**) Minimal suppression of pS6 and pY-STAT3 by AQ (30μM × 3 h) when administered prior to stimulation with media conditioned by HS-5 stromal cells in representative patient sample 10. (**E**) AQ did not alter expression of gp130 in representative patient sample 10 after exposure to 25 μM AQ × 3 h. (**F**) Composite graphs of primary patient samples supported by HS-5 stromal cells in transwell co-culture characterized as non-responders (*n* = 4, *p* = 0.1758) or responders (*n* = 16, *p* < 0.0001). Non-responders were defined as having an average AQ vs. vehicle control difference in Annexin V + cells of <15% across all time points and responders >15%. (**G**) Composite graph of the OCRs of non-responders (*n* = 2) with a *p* value of 0.8921, and composite graph of the OCR of all responders (*n* = 15) with a *p* value of 0.001 (**H**) Table with mutations for each primary cell sample classified as responder vs. non-responder. Individual Unique Patient Numbers (UPNs) are listed in brackets. Oligo = oligomycin, FCCP = carbonyl cyanide-4-(trifluromethoxy) phenylhydrazone, and Ant/Ro t= antimycin A/rotenone, and AQ = atovaquone.

**Table 1 cancers-15-01344-t001:** Patient demographic and correlative biology studies.

Characteristic, *n*(%)		All Patients (*n* = 26)
Age, y	Median (range)	12 (0.6–19.7)
+Blast BM, %	Median (range)	68.5 (0–97)
Sex	Female	12 (46)
	Male	14 (54)
Race	White	21 (81)
	Black or African American	4 (15)
	Asian	1 (4)
Ethnicity	Hispanic	8 (31)
	None-Hispanic	18 (69)
French-American-British Classification	M1/M2	11 (42)
	M4/M5	9 (35)
	M4eo	1 (4)
	M7	4 (15)
	NA	1 (4)
CNS Disease	Yes	15 (58)
Relapse	Yes	10 (38)
EOI1_MRD	Positive	9 (35)
	Negative	17 (65)
Clinical Outcome	Alive	19 (23)
	Death from disease	4 (15)
	Treatment related mortality	3 (12)
High Risk by 1831	Yes	12 (46)
KMT2A rearranged	Yes	5 (19)
FLT3-ITD (allelic ratio≥0.1)	Yes	6 (23)
IDH1 or IDH2 mutaion	Yes	5 (19)
Annexin	Yes	20 (77)
In Vitro AQ Response	Responder	16 (80)
	Non-Responder	4 (20)
Oxygen Consumption Rate Evaluation	Yes	17 (65)
Phosphoflow and Surface Flow Cytometry Evaluation	Yes	18 (69)

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
