# Peer review of "Repurposing Atovaquone as a Therapeutic against Acute Myeloid Leukemia (AML): Combination with Conventional Chemotherapy Is Feasible and Well Tolerated"

_cancers, 2023, doi:10.3390/cancers15041344_

Round 1

Reviewer 1 Report

In this manuscript, the authors report on a clinical trial incorporating the anti-microbial drug atovaquone in the treatment of children with AML, based on reports of anti-leukemic activity of atovaquone.  They provide extensive data on tolerability, safety, and pharmacokinetics of atovaquone in this study, as well as on associated ex vivo studies of atovaquone on samples from some of the patients.

Overall, this is a very well-executed study, that provides a wealth of clinically relevant data on the use of atovaquone in this patient population.  As such, it will help other investigators seeking to leverage this approach.  In addition, the following points should be considered:

1.  In the Discussion (starting at line 462), the authors refer to their data showing a lower concentration of atovaquone than anticipated from the literature.  They then write, “These findings are not surprising given the long half-life of AQ, poor bioavailability (particularly given dependency of absorption on a high-fat diet), and delayed onset of initiation of therapy (day 6).”  However, the data shown in Figure 1E do not show much of a trend for increasing concentration with time, suggesting that starting on day 6 and the long half life (presumably relating to a longer time to reach steady state) were not relevant issues.  It would be worthwhile to more critically compare these data with the published literature on atovaquone pharmacokinetics to more rigorously address the issue of this discrepancy in achieved concentrations.

2.  Later in the same paragraph, the authors discuss target concentrations of atovaquone, and refer to data from therapeutic studies.  The concentration needed for prophylaxis for most antimicrobials is significantly lower (and not necessarily continuous, as exemplified with trimethoprim-sulfamethoxazole regimens) than for therapeutic use.  Therefore, this section should dissect the issue of concentrations necessary for therapeutic versus prophylactic use of this drug more clearly.

3.  In the final paragraph of the Discussion, the authors indicate that a randomized control trial of atovaquone is likely impractical for an uncommon disease such as pediatric AML.  They then go on to say that it is “unnecessary” (lines 507-508).  It would seem that there should be more caution with a statement like that.  First, there are many examples from the oncology literature in which addition of a drug that has anti-cancer activity to an established regimen actually has a deleterious effect.  Second, since atovaquone does not appear to be as effective as trimethoprim-sulfamethoxazole for PJP prophylaxis, switching prophylactic regimens could have detrimental effects.  The authors might refine how they frame the issue of evaluating whether incorporation of atovaquone is truly beneficial to these patients.

4.  A number of groups (and maybe even this one) have shown that atovaquone decreases STAT3 phosphorylation in a variety of cancer cell types.  From the flow cytometry presented (Figure 5D and supplementary data), it is not clear that there was much STAT3 phosphorylation in these ex vivo cultures in the absence of the conditioned media.  Was STAT3 phosphorylation assessed in any of the bone marrow biopsy samples by immunohistochemistry to assess whether STAT3 is phosphorylated in the myeloblasts in the patients?  The authors may want to comment further on the negative finding with respect to STAT3 given the countervailing published studies.

Reviewer 2 Report

In this interesting work McLean and collegues show the positive effect of the use of Atovaquone as agent for the required prophylaxis against pneumocystis jiroveci pneumonia during conventional chemotherapy. Atovaquone has known to have antileukemic effect by inhibition of mitochondrial OXPHOS and here they demonstrate it to be well tolerated, well accepted and able to reach antileukemic concentration in patients plasma.

Figure 4- Since Atovaquone alone does not prolong much PDX survival and since in my experience hCD45 positivity in mice blood does not faitfully reflect tumor burden in the bone marrow it would be more informative to assess hCD45 positivity in IHC in the PDX femur and spleen or easyer by FACS in PDX isolated bone marrow cells.

Figure 5c 5d 5e SF12- Since cell mortality can induce unspecific positivity to antibodies it would be interesting to see the viability gate for the analyzed cells.
